# Spatial Pattern Analysis of Xinjiang Tourism Resources Based on Electronic Map Points of Interest

**DOI:** 10.3390/ijerph19137666

**Published:** 2022-06-23

**Authors:** Yao Chang, Dongbing Li, Zibibula Simayi, Shengtian Yang, Maliyamuguli Abulimiti, Yiwei Ren

**Affiliations:** 1College of Geography and Remote Sensing Sciences, Xinjiang University, Urumqi 830046, China; cyydfck@stu.xju.edu.cn (Y.C.); lidongbing@stu.xju.edu.cn (D.L.); maly1106@163.com (M.A.); 2Key Laboratory of Wisdom City and Environment Modeling of Higher Education Institute, Urumqi 830046, China; 3Institute of Water Science, Beijing Normal University, Beijing 100875, China; yangshengtian@bnu.edu.cn; 4College of Resources and Environmental Engineering, Ludong University, Yantai 264025, China; ryw199801@163.com

**Keywords:** Xinjiang, tourism resource distribution, spatial analysis, points of interest

## Abstract

This study considers the Point of Interest data of tourism resources in Xinjiang and studies their spatial distribution by combining geospatial analysis methods, such as the average nearest neighbor index, standard deviation ellipse, kernel density analysis, and hotspot analysis, to explore their spatial distribution characteristics. Based on the analysis results, the following conclusions are made. Different categories of tourism resource sites have different spatial distributions, and all categories of tourism resources in Xinjiang are clustered in Urumqi city. The geological landscape resource sites are widely distributed and have a ring-shaped distribution in the desert area of southern Xinjiang. The biological landscape resources are distributed in a strip along the Tianshan Mountains. The water landscape resources are concentrated in the northern Xinjiang area. The site ruins are mostly distributed in the western region of Xinjiang. The distributions of the architectural landscape and entertainment and shopping resources are highly coupled with the distribution of cities. The distributions of the six categories of tourism resource points are in the northeast-southwest direction. The centripetal force and directional nature of the resource points of the water landscape are not obvious. The remaining five categories of resource points have their own characteristics. The distribution of resources in the site ruins is relatively even, and there are many hotspot areas in the geomantic and architectural landscapes, which are mainly concentrated in Bazhou and other places. The biological landscape has many cold-spot areas, distributed in areas such as Altai in northern Xinjiang and Hotan in southern Xinjiang. The remaining four categories have cold-spot and hotspot areas with different distributions. Tourism is an important thrust for economic development. The study of the distribution of tourism resources on the spatial distribution of tourism resources has clear guidance for later tourism development, can help the tourism industry optimize the layout of resources, and can promote tourism resources to achieve maximum benefits. The government can implement effective control and governance.

## 1. Introduction

### 1.1. Theoretical Background

Tourism resources, as an important spatial carrier of tourism reception and the basic conditions of tourism development, its location, and distribution, play a key role in the rational development of tourism resources, regional tourism market competition, and industrial upgrading and innovation. With the increasing share of tertiary industry in the national economic structure of China, tourism has gradually become a pillar industry [1]. In 2019, China received a total of about 6 billion domestic tourists and 145 million inbound tourists, with the total tourism spending reaching CNY 5,725,092 billion. In recent years, the tourism index has shown a general upward trend [2]. In this context, Xinjiang has positioned tourism as an important strategic pillar industry, and the tourism industry in Xinjiang has been transformed from a large tourism resource area to a strong tourism economy area. The tourism industry in Xinjiang has shown rapid development, which has led to the optimization of the spatial economic structure of the city and strengthened various economic activities of the spatial elements of the city [3].

From 2015 to the present, the concept of holistic tourism has gradually matured in China, which is a kind of tourism by which the advantageous industry, through the economic and social resources in the region—especially tourism resources, related industries, the ecological environment, public services, institutional mechanisms, policies and regulations, civilization quality, etc.—carries out all-round and systematic optimization and enhancement in order to achieve the organic integration of regional resources, industrial integration and development, social construction and sharing, and drive and promote the coordinated economic and social development of a new regional coordinated development concept and model [4]. Tourism resources represent the premise of tourism development and the industrial substrate of tourism. The spatial concentration and dispersion of tourism resources are important factors affecting tourism activities [5], and their reasonable spatial distribution is of high significance to the integrated development of tourism under the concept of a holistic tourism destination. The development and research of the tourism resource distribution play a key role in the aggregation of tourism industries and the development of backward industries [6].

### 1.2. Literature Review

The study of the spatial distribution of tourism resources can provide a practical value basis for the optimization and upgrading of regional tourism resources, which are the “material” of tourism, the core content of tourism, and the core element of forming attraction [7]. The previous studies on the tourism resource distribution have mainly relied on statistical data [8], social surveys, and interview consultations and have combined the traditional geographic theories, such as spatial structure theory and location theory, to conduct qualitative research. However, this approach has many shortcomings [9]. With the development of geographic information technology, methods such as GIS spatial analysis and Geodetector [10] have become the main means for studying the distribution, evolution, and influencing factors of resources. Fang et al. [11] explored the spatial distribution pattern of ski resorts in China using the Geodetector and GIS spatial analysis techniques. The studies on spatial distribution mostly used kernel density analysis, the imbalance index, spatial autocorrelation, complex network analysis, and geographic linkage rate quantitative indicators and methods [12,13]. Dolores et al. [14] used the GIS spatial analysis methods to study the spatial distribution of tourism supply in Andalusia. Lu [15] used traditional geographic information methods to analyze the spatial distribution of residential-category scenic spots in Shanxi. Hu et al. [16] used Google Earth and ArcGIS to study the spatial and temporal evolution characteristics of scenic spots AAA and above in Shanxi Province, China.

With the increase in big geographic data applications, such as big migration data [17], cell phone signaling data [18], POI (Point of Interest) data [19], public transportation swipe cards, and social media, geography has ushered in changes in research methodology [20], combining the real-time, fast, and efficient characteristics of big data with a more refined research scale in geomatics. Park et al. [21] used big data mining methods to study the spatial structure of tourism destinations. Pei et al. [22] made full use of big data thinking—the distribution, change, and mechanism of the study object were revealed by the ideas of correlation, synthesis, and inversion from the rich big footprint data. In terms of spatial distribution characteristics, related studies focused on using the POI data to explore how various service industry facilities are distributed spatially [23]. Dimelli et al. [24] studied the economic effects of tourism in the Greek island as a center of research. Han et al. [25] studied the distribution characteristics of new coronavirus flows in Beijing using POI data with population flow attribute data. Miao et al. [26] developed a method that can automatically identify urban space and urban functional areas using a combination of spatial information created by users in microblogs and GPS data. Krataithong et al. [27,28] used the combination of the cab’s positioning system and big geographic data, such as tourism location data and social media data, to analyze urban tourism patterns, mostly using the city area as a research scale. Meanwhile, small-scale tourism geography studies, which used big data and regarded villages and small towns as research objects, have also emerged under the rural revitalization strategy [29]. The POI is a term used to refer to geographic data points in GIS, but unlike traditional official data, POI data include information on location characteristics [30]. Thus, by using POI data, accuracy can be improved, and the existence of features can be reflected in real time. Recently, there have been several studies on the distribution of tourism resources using POI data at the provincial scale in China. Combining geospatial analysis with big data methods provides a mechanism for leveraging and maximizing traditional survey data to further existing efforts in a meaningful way [31].

Since the outbreak of the COVID-19 pandemic, China’s tourism industry has been showing a temporary decreasing trend. It may be worth mentioning that government travel restrictions imposed during the pandemic have affected tourism. Many tourists are choosing not to travel, but many who still wish to travel must abide by the travel restrictions of their country. Yang et al. [32] showed that the tourism and restaurant industries suffered more losses than the entertainment and digital office industries and that there were different stages of virus transmission such as outbreak and fading. In Xinjiang, which is the largest province in China in terms of area, people have been avoiding long traveling to prevent virus transmission, so a local trip has become a good choice. However, by applying efficient policy measures, the epidemic has been stabilized in China, and the spread of the epidemic has been successfully controlled [33]. This helped to recover the tourism industry, which has contributed to the economic recovery. The judgment of the current distribution pattern of tourism resources helps the government and tourism companies to develop the best model for the current form of tourism. Therefore, it is meaningful and necessary to study the distribution pattern of tourism resources. This paper takes Xinjiang as the study area and explores the spatial distribution characteristics of Xinjiang tourism resources. This paper differs from other pieces of literature in that (1) the POI data of Xinjiang tourism resources are used, and their spatial distribution characteristics are analyzed. (2) The future outlook of tourism resources development in the context of the epidemic is discussed, and it is hoped to promote the finding of a new model of tourism.

## 2. Materials and Methods

### 2.1. Study Area

Xinjiang (73°40′–96°18′ E, 34°25′–48°10′ N) is located in the semi-arid region of northwest China, the hinterland of Asia and Europe, bordering Inner Mongolia, Gansu, Qinghai, and Tibet and also eight countries externally. The land border reaches more than 5600 km. The climate of Xinjiang is temperate continental, with basins and mountain ranges distributed among each other. Xinjiang has a long history and is rich in natural and human tourism resources. The Chinese Ministry of Culture and Tourism classifies scenic spots, from low to high, as 1A, 2A, 3A, 4A, and 5A. According to the Xinjiang Department of Culture and Tourism (http://wlt.xinjiang.gov.cn/, accessed on 20 May 2021), since May 2021, the autonomous region has had 14 5A tourist attractions and 107 4A tourist attractions. The research scope of this paper includes the prefecture-level administrative units in Xinjiang, among which the research data of Ili include only the county-level administrative regions in Ili (Figure 1).

### 2.2. Data Source and Processing

In this paper, each state and city in Xinjiang is studied, and the required POI data are mined. The POI data have the advantages of a large data volume, comprehensive coverage, high accuracy, real-time reflection of the existing state of features, and easy access and thus have been widely used in statistical studies, social surveys, and other types of research. The POI data include eight attributes: name, category, address, longitude, latitude, administrative area, address, and administrative district number [34]. In this study, the Bigemap map downloader (http://www.bigemap.com/, accessed on 18 June 2021) [35] and Python web crawler [36] were used to query the keywords “tourism,” “scenic spot,” and “landscape” on the Gaode Map open platform (https://lbs.amap.com, accessed on 18 June 2021) to obtain the data from June 2021. The acquired POI data were pre-processed to remove duplicate and irrelevant data, and the remaining valid data were classified. The valid data included 2139 samples. The classification was performed based on the classification criteria in “*Classification, Survey and Evaluation of Tourism Resources* (GB/T 18972-2017)” (http://openstd.samr.gov.cn, accessed on 29 December 2017) and the special characteristics of Xinjiang tourism resources [37]. The POI data of Xinjiang tourism resources were divided into six categories, as shown in Table 1 and Figure 1. The data on the number and distribution of national scenic spots, tourism revenue, and the number of visitors in different years were obtained from the Xinjiang Department of Culture and Tourism (www.xinjiangtour.gov.cn, accessed on 1 March 2022). In addition, the vector boundaries of the administrative regions in Xinjiang used in this study were obtained from the latest National Basic Geographic Information System database (www.ngcc.cn, accessed on 8 October 2020).

### 2.3. Methods

#### 2.3.1. Average Nearest Neighbor Index

The average nearest neighbor index is an effective spatial measure for quantitatively describing the proximity of spatial point-like elements and judging their spatial pattern special and distribution patterns [38], and it is calculated by:(1)$$ANN={\bar{D}}_{0}/{\bar{D}}_{E}$$
(2)$${\bar{D}}_{0}=\sum_{i=1}^{n}d_{i}/n$$
(3)$${\bar{D}}_{E}=\frac{1}{2}/\sqrt{n/A}$$
where *ANN* denotes the average nearest neighbor index; $\bar{D}$_0_ denotes the average observed distance between an element and its nearest neighboring element; $\bar{D}$*_E_* denotes the expected average distance between elements in a random pattern; *n* is the number of elements; *A* is the area of the study area.

When, *ANN* < 1, the element distribution tends to be agglomerative, and the smaller the *ANN* value is, the higher the degree of agglomeration will be; when *ANN* = 1, the element distribution tends to be random; when *ANN* > 1, the element distribution tends to be discrete, and the larger the *ANN* value is, the higher the degree of dispersion will be.

#### 2.3.2. Standard Deviation Ellipse Analysis

The direction, trend, distribution characteristics, and potential correlation with specific elements of data distribution can be identified by the standard deviation ellipse analysis [39]. In the standard deviation ellipse analysis, the long half-axis of an ellipse indicates the direction of data distribution, while the short half-axis indicates the centripetal nature of the data and the distribution range. The larger the difference between the long and short half-axes is, the more pronounced the directionality of the data distribution will be. The mathematical expression of the standard deviation ellipse analysis is as follows:(4)$$E_{x}=\sqrt{\frac{\sum_{i=1}^{n}\left(x_{i}-\bar{X}\right)}{n}}$$
(5)$$E_{y}=\sqrt{\frac{\sum_{i=1}^{n}\left(y_{i}-\bar{Y}\right)}{n}}$$
(6)$$\tan\theta =\frac{{\left(\sum_{i=1}^{n}a_{i}{}^{2}-\sum_{i=1}^{n}b_{i}{}^{2}\right)}^{2}+\sqrt{{\left(\sum_{i=1}^{n}a_{i}{}^{2}-\sum_{i=1}^{n}b_{i}{}^{2}\right)}^{2}+4{\left(\sum_{i=1}^{n}a_{i}b_{i}\right)}^{2}}}{2\sum_{i=1}^{n}a_{i}b_{i}}$$
where *E_x_* and *E_y_* denote the long and short axes of a standard ellipse, respectively; *x_i_* and *y_i_* denote the coordinates of an element *i*; $\bar{X}$ and $\bar{Y}$ represent the mean centers of all the elements; *n* is the number of elements; *θ* is the rotation angle of the standard ellipse; *a_i_* and *b_i_* represent the distances from an element *i* to the mean center in the directions of the long and short axes, respectively.

#### 2.3.3. Nuclear Density Analysis

In the analysis of the spatial clustering characteristics of data points, kernel density analysis reflects the relative concentration of the spatial distribution of data points [40] and can be used to study the spatial distribution characteristics of data from the aspects of data samples. A larger kernel density value indicates a denser distribution, which is calculated by:(7)$$f(x)=1/nh\sum_{i=1}^{n}k(\frac{x-x_{i}}{h})$$
where *f (x)* denotes the kernel density function; *k* is the kernel function; *x_i_* is the specific location in the space of the tourism resource points in the region formed by *x* as the center of a circle; (*x* − *x_i_*) denotes the distance from the estimated point to the sample; *h* is the radius; *n* denotes the number of sample points.

#### 2.3.4. Hotspot Analysis

The standard deviation ellipse analysis is mainly used to analyze the distribution direction and trend of various categories of tourist attractions in the whole area of Xinjiang, but it cannot reflect the degree of clustering of tourist resources in cities in Xinjiang. Therefore, the cold-spot and hotspot analysis method [41] is employed to analyze the clustering of six categories of tourist attractions in various states of Xinjiang. This method—namely, *Getis-Ord Gi** statistics—uses ArcGIS10.4 software to examine whether there is spatial clustering in local distribution, with *Z* scores and *p* values in the high- and low-value areas of agglomeration. In this study, the numbers of tourism resource points in each city and state of the six categories were input to this method to obtain and visually express the distribution of the cold-spots and hotspots of various categories of scenic touristic spots in each city of Xinjiang.

## 3. Spatial Distribution Characteristics of Different Categories of Tourism Resources

### 3.1. Quantitative Analysis

Xinjiang, as a major tourism province in China, has many different types of tourism resources [42]. Under the precise prevention and control of COVID-19, the Xinjiang Autonomous Region received a total of 158 million tourists and achieved a tourism revenue of CNY 99.212 billion in 2020. From January 2021 to November 2021, the autonomous region received a total of 180 million tourists and a achieved tourism revenue of CNY 135.026 billion, which represents an increase of up to 51.67% compared to the previous year.

According to Table 2, various categories of tourism resources are distributed in all of the cities in Xinjiang, among which Urumqi has the largest number of tourism resource sites (373), while Bozhou has the smallest number of tourism resource sites (23). Specifically, the architectural landscape has the largest number of resource points, with a total of 1145 resource points, ranking first among the categories of resource points; Urumqi included the most resource points among all cities, having 283 of them. It is followed by Changji, which is inextricably linked to the region’s long cultural history, and its position can be regarded as the center of gravity of the regional economy [43]. In recent years, the output value of the tertiary industry in Xinjiang has gradually increased; the government has given importance to the tourism and cultural industry and has vigorously developed tourism resources. In addition, historical and ancient buildings have been better protected, and a number of new architectural landscape tourism resources have been developed, such as private museums, street parks, urban food streets, and specialty shopping malls.

The site ruins rank second in terms of the number of tourism resource sites among the six categories, with a total of 464 sites. Among all cities, Bazhou has the most tourism resource sites, namely, 98 sites. Bazhou has a long history, with thousands of years of development. Both the South and Middle Silk Roads pass through Bazhou, which provides unique tourism resources. The state has various categories of tourism resources and numerous historical sites and humanistic landscapes, such as the Iron Gate Pass, the ruins of the ancient city of Loulan, the Sikqin Thousand Buddha Cave, and the Milan Ruins.

There are a total of 278 geological landscape resources in Xinjiang. The geomorphic landscape resource sites are mainly located in the north of Xinjiang, among which the Altai region has 47 sites, followed by Bazhou and Ili. The reason for this is that northern Xinjiang has more geomorphic categories than southern Xinjiang, including deserts, mountains, valleys, oases, and other topographical features. Accordingly, northern Xinjiang includes world-famous geomorphological tourism resources, such as World Devil City, Tianshan Grand Canyon, Karamay Black Oil Mountain, and Kumutag Desert.

Furthermore, there are 93 biological landscapes and 121 water landscapes in Xinjiang. Such resource sites have been developed and constructed due to the local natural conditions; there are many lakes in Xinjiang, with a total area of 9700 square kilometers, accounting for more than 0.6% of the total area of Xinjiang. In recent years, Xinjiang has been paying more attention to ecological protection, and lakes and rivers have been protected. Biological landscape resource sites and water landscape are related, and wetlands are protected; the attraction of migration and the settlement of different species of organisms are ensured, forming a seasonal biological landscape.

The entertainment and shopping resources have a total of 60 sites, the least among all categories; Urumqi has the most sites—10 sites exactly. Still, this category has fewer tourism resources compared to regional centers such as eastern and central China, reflecting the lack of innovative development of Xinjiang’s current tourism industry, which still strongly relies on the original tourism resource endowment.

### 3.2. Distribution Pattern of Tourism Resources

According to Equation (1), the average nearest neighbor index (*ANN*) results of the six categories of tourism resources in Xinjiang show that *ANN* is less than one, and the *p*- and *z*-values are less than −2.58, which passes the significance test at the confidence level of 0.01; thus, the original hypothesis is rejected with 99% certainty. Therefore, it can be judged that all categories of tourism resources in Xinjiang are a clustered distribution. By overlaying the kernel density layer and vector map, it is found that the tourism resources in Xinjiang are mainly concentrated in Urumqi city, as shown in Figure 2, and decrease in all directions from Urumqi as a center, as shown in Table 3.

### 3.3. Density Analysis of Tourism Resources

The kernel density analysis function (Equation (7)) of ArcGIS10.4 is used to analyze the distribution patterns of different categories of tourism resources in Xinjiang, and then the kernel density distribution maps of the six categories of resource points are obtained. As shown in Figure 3, Urumqi is a high-density distribution area for various categories of tourism resource points, and the distribution of each category of resource points has its own characteristics in terms of distribution.

(1)Geological landscape resource sites (Figure 3a) are tourism sites with mountains, valleys, and typical geological zones as the main categories, forming high-density clusters in Ili, Altay, Tacheng, and Karamay and showing roughly triangular clustering characteristics in northern Xinjiang. This is due to the distribution of multiple categories of geological landforms in northern Xinjiangl; the Ili River valley opens to the west, receiving water vapor from the Atlantic Ocean, but there is a blockage by the Tianshan Mountains, trapping water and gas, making sufficient moisture, and forming a vast oasis valley [44]. The climate of Karamay is arid, with little rain and high winds during the year, forming a typical tourist attraction of the Yardan landform—the World Devil City. A high-density area is formed in the north of the country, while the south of the country includes a catchment area in the Kashgar region. Kashgar is rich in tourism resources, with the Pamir Plateau in the west, the Taklamakan Desert in the east, and the Karakorum Mountains in the south, and it has the 8611-meter-tall Chogori Peak on its territory.(2)The resource sites in the architectural landscape (Figure 3b) are the same as those in the category of entertainment and shopping; they are mostly concentrated around cities, they are closely related to the distribution of cities, and they overlap with the distribution of famous scenic spots, such as the ancient city of Kashgar, the Forbidden City of Jiaohe, the ancient city of Loulan, and the ancient city of Gaochang.(3)Biological landscape resource sites (Figure 3c) are located in a large area in the north of the Tianshan Mountains and are concentrated in high densities in Urumqi, Changji, and Ili on the northern slopes of the Tianshan Mountains. These areas include many different categories of topographic features in Xinjiang, which are suitable for the migration and settlement of organisms. Additionally, most of the nature reserves in Xinjiang are distributed in the north of the Tianshan Mountains; the newly established Altai Kekesu Wetland and the hot spring Xinjiang Northern Salamander National Nature Reserve are located in the northern region. A good natural environment provides all the necessary prerequisites for the formation of biological landscape tourism resources [45].(4)The resource points of the water landscape (Figure 3d) form a high-density area near the famous lake scenic spots in Xinjiang (e.g., Bosten Lake, Selim Lake, and Aibi Lake), concentrated in Bazhou, Urumqi, Altay, and Ili. Bazhou’s Bosten Lake is the largest inland freshwater throughput lake in China and a national 5A tourist attraction. It is located in the south of the Tianshan Mountain. The Altai region, with its many mountains, is conducive to the collection of water sources and forms famous water landscapes, including the Ertzis River, Kanas Lake, and Ulungu Lake.(5)The distribution of entertainment and shopping resources (Figure 3e) is highly coupled with the distribution of cities in Xinjiang, and the development of the tourism industry of the recreation and shopping category depends on the city economy [46]. There is no obvious high-density area in Xinjiang but only a small amount of concentrated distribution in cities along the north–south railroad line, such as Urumqi, Changji, Kulle, and Kashgar.(6)The resources in the category of site ruins (Figure 3f) are widely distributed throughout Xinjiang, with areas such as Urumqi and Turpan as a core, which are inseparable from the long history and culture of Xinjiang. The high-density areas are mostly located along the route from Hami to Ili and from Altay to Kashgar, which approximately coincides with the ancient Silk Road in Xinjiang. Due to the extensive distribution of deserts and the long history, the site ruins show a circular distribution with a low degree of concentration in the southern Xinjiang region.

### 3.4. Direction and Trend of Tourism Resource Distribution

The standard deviation ellipse analysis (Equations (4)–(6)) in the ArcGIS10.4 software was used to analyze the six categories of tourism resources, and the directionality and trend of their distribution were obtained. According to the distribution status in Figure 3, the distribution of the six categories of tourism resources in Xinjiang shows a northeast-southwest direction, which coincides with the direction of the whole area of Xinjiang. This is because the southeast of Xinjiang is mostly restricted by natural factors, such as deserts and Gobi, so the tourism resources are mostly concentrated in the northwest of Xinjiang.

The length of the *X*-axis indicates the resource distribution and obvious centripetal force presented by the distribution (Table 4); a longer *X*-axis indicates a larger distribution range of tourism resources and a less obvious centripetal force—and vice versa. The longest *X*-axis corresponds to the resource points of the water landscape, followed by the geological landscape, site ruins, architectural landscape, biological landscape, and entertainment and shopping, so it can be concluded that the resource points of the water landscape are widely distributed in the whole region and that the centripetal force is not obvious. In contrast, the resource points in the category of entertainment and shopping show a high degree of concentration and have a small distribution range. Additionally, the centripetal force is obvious, as shown in Figure 4.

The length of the *Y*-axis indicates the distribution direction of the data; the distributions of all types of tourism resource points in Xinjiang show a northeast-southwest direction, which is roughly consistent with the shape of Xinjiang. Among the six categories, the category of entertainment and shopping has the most obvious distribution direction of resource points, while that of the biological landscape is the least obvious; the remaining four categories are between these two categories.

In terms of the deflection angle of the ellipse, the list of categories in descending order is as follows: biological landscape, entertainment and shopping, architectural landscape, geological landscape, heritage sites, and water landscape. Namely, the deflection angle of the biological landscape is the largest, and it equals 81.578°, which roughly coincides with the orientation of the Tianshan Mountains in Xinjiang and confirms the distribution of the biological landscape resource points (Figure 3c). The deflection angle of the water landscape is the smallest, and its distribution direction roughly coincides with the straight-line distance from the Altay region to the Kashgar region, along which there is an extensive distribution of oases. The topography of Xinjiang, as “three mountains sandwiched by two basins”, is longitudinal, with natural conditions for the formation of rivers and lakes, and the major rivers and lakes in Xinjiang, such as the Manas River, are concentrated around the route lakes.

### 3.5. Distribution Characteristics of Tourism Resource Categories in Cities

To clarify the degree of upper concentration of various categories of tourism resource points at the city level, ArcGIS10.4 was used to conduct the hotspot analysis (Getis-Ord Gi*) in order to extract the hotspot and cold-spot areas of different categories of resource distribution. The results are presented in Figure 5, where the yellow color indicates insignificant areas. The hotspot areas are denoted by a red color, the cold-spot areas are denoted by a blue color, and the color darkness indicates the size of the resource point concentration.

As shown in Figure 5, Urumqi is mostly a hotspot or a sub-hotspot in the six categories of tourism resources, especially in the categories of architectural landscape, entertainment and shopping, and biological landscape. The architectural landscape category has a circular hotspot around Urumqi, showing the city’s ability to develop, cluster, and protect tourism resources as a regional economic and cultural center [47]. Bazhou is a hotspot in the categories of geomorphic landscape, water landscape, and site ruins. In addition, the geomantic landscape has hotspots in the Altai region, the city of Karamay, and the Ili Prefecture, and it has cold-spots in places such as Bozhou, whose distribution is roughly similar to that of the site ruins. Bozhou is located in the western part of Xinjiang and is adjacent to the Tianshan Mountains, but the typical scenic areas of the Tianshan Mountains have been well developed in places such as Changji and Urumqi [48], forming national 5A-grade scenic areas, such as Tianshan Grand Canyon and Tianshan Tianchi scenic areas, and lacking the attractiveness of secondary development.

The cold-spot areas of the biological landscape are more distributed than other categories—mostly in the western regions of Xinjiang, such as Ili and Tacheng—and the hotspot areas are concentrated in the middle part of Tianshan Mountain due to the natural conditions. The entertainment and shopping resources have a significant hotspot area in Urumqi; Hotan and Kashgar are also regarded as sub-hotspot areas, but their degree is not high. The cold-spot areas of the water landscape resources are mainly concentrated in Kezhou and Aksu, while Kashgar and Hotan are insignificant areas. This is because the climate of southern Xinjiang is arid and is dominated by desert distribution, and evaporation is greater than precipitation, so it is difficult to form surface runoff. Additionally, the Tarim River is a seasonal river, so it is not easy to develop tourism resources of the water landscape category.

## 4. Influencing Factors of the Spatial Distribution of Tourism Resources

The factors influencing the spatial distribution of different categories of tourist resources in Xinjiang are divided into natural geographic factors and socio-economic factors [32]. The natural geographic factors provide the basis for the formation and development of various categories of tourist attractions. Meanwhile, with the advent of the era of mass tourism, socio-economic factors have become an important driving force for the development of various categories of tourist attractions [33], creating different distribution characteristics of tourist resource categories.

### 4.1. Natural Geographical Factors

Xinjiang is a vast area, occupying one-sixth of the overall land area of China. It is surrounded by mountain ranges and basins, with the Altai Mountains in the north, the Kunlun Mountains in the south, separated from Tibet, and the Tianshan Mountains in the middle, between which the Tarim Basin and Junggar Basin are located. The climate in Xinjiang is arid and windy, which is beneficial for unique geological and geomorphological categories.

The economic development of southern Xinjiang is not coordinated with that of northern Xinjiang [49], and its development ability of natural tourism resources such as the geomantic landscape is weak due to factors such as a small population in southern Xinjiang. Therefore, tourism resources are mainly distributed in northern Xinjiang. There are many rivers and lakes in Xinjiang, and their distribution is highly coupled with the tourism resources of the water landscape, forming a concentrated distribution of tourism resources around the famous lakes, such as Saarim Lake and Bosten Lake, and the water landscape of the city of Urumqi. The distribution of lakes and water systems, the barrier of tall mountains, and the construction of ecological protection forests and other natural factors, coupled with the ecological protection of the environment, have become the priorities for the government in recent years [50], and the clustering of biological landscape tourism resources has been promoted.

The flat terrain and water sources are necessary for human survival, and the oasis on the vast desert has become a unique geomorphic resource in Xinjiang, distributed in stripes along rivers or near lakes. In the high mountain foothills on the edge of the Tarim Basin and Junggar Basin, the distribution of oases provides the natural foundation for heritage sites and the tourism resources of architecture and facilities. Many heritage sites and tourism resources of architecture and facilities, such as the ancient city of Kashgar and the ancient city of Gaochang, are distributed on the oasis.

### 4.2. Socio-Economic Factors

Xinjiang has a long history, with multi-ethnic populations and a multi-cultural convergence. In recent years, with the construction of the overland Silk Road Economic Belt, Xinjiang has once again brought cultural and economic development to life [51]. According to the Xinjiang Annual Statistical Report (http://tjj.xinjiang.gov.cn/, accessed on 1 March 2022), from 2011 to 2021, Xinjiang’s tourism industry achieved a tourism revenue of CNY 363.258 billion and received about 213 million domestic and foreign tourists by 2019 (Figure 6). In 2020, tourism revenue and reception decreased due to the impact of the COVID-19 pandemic [32], so the Xinjiang Department of Culture and Tourism specifically launched the “*Ten Measures for Epidemic Prevention and Control to Promote the Recovery and Development of the Tourism Market under the Epidemic Normalization”* to promote the rapid recovery of the tourism industry. According to recent reports, the year 2021 has already surpassed the year 2020 in terms of revenue and receipts in the first 11 months, and the tourism fever in Xinjiang has re-emerged under the tight control of the COVID-19 pandemic [52].

The government policies and importance have been of guiding significance to the formation and development of scenic touristic spots [53]. The Xinjiang Winter Tourism Promotion Conference 2020 was held in Beijing, Shanghai, Hangzhou, and Chengdu, where all relevant cities in Xinjiang introduced a series of tourism lines and supporting incentives. For instance, Ili and other cities introduced free entrance fees for national tourists to scenic spots above the A level and tourism transportation subsidies for tourists from other provinces. Since the 14th Five-Year Plan, the national strategy of rural revitalization has led to the development of regional specialties such as safflower, Chinese wolfberry, pomegranate, and tomato, processing into agrotourism and the synergistic development of agriculture and tourism [54], which has enriched tourism resources such as construction facilities. Meanwhile, in the context of China becoming the world’s largest carbon emitter and promoting the “double carbon” goal, the adjustment of the industrial structure, industrial integration, and the full development of tourism and other low-carbon industries have become the main development direction [55].

Recent urbanization development has also brought opportunities for tourism resources in Xinjiang, with urbanization driving the development of urban tourism, providing the most obvious benefits to the category of entertainment and shopping tourism resources. The construction of the Urumqi metropolitan area has shortened the intercity commuting time [56], allowing for more frequent travel between cities and an increase in short-term tourist categories such as half- and one-day trips, counteracting the further improvement of urban service facilities and quality. In July 2021, Xinjiang’s first theme park opened in Urumqi, and in August 2021, the largest water park in southern Xinjiang, named the Hotan Happy Water World, opened and started operating.

## 5. Discussion and Recommendations

The future development direction of the Xinjiang tourism industry should follow the perspective of the development of holistic tourism destinations. The government has been guiding the strengths and weaknesses of the municipalities to create regional characteristics, pointing out that cultural industries lead the way in tourism development. Since the outbreak of the COVID-19 pandemic, tourism has suffered significant losses worldwide. Xinjiang’s tourism industry should seize the opportunity of the development of “One Belt and One Road” and adopt the strategy of driving the development of the Xinjiang tourism industry with a competitive effect, focusing on highlighting the advantages of Xinjiang, creating brands with cultural characteristics, deepening the cultural connotations of tourist attractions, and accelerating the transformation from homogenization to branding. Additionally, the tourism industry could be improved by establishing the tourism logo of “Xinjiang is a good place”, combining new media and We Media to expand the influence of Xinjiang tourism on the mainland and even abroad and promoting the development of different forms of rural, ecological, and folklore tourism in tandem with a unique culture.

In addition, in the comprehensive construction of a well-off society and China’s precise epidemic prevention and control, tourism flourishing gradually revived, and innovative tourism has become the way to develop the tourism industry in Xinjiang. The traditional six elements of tourism—namely, food, accommodation, travel, tourism, shopping, and entertainment—should be combined with new tourism elements, including business, recreation, learning, leisure, fun, and curiosity. Taking advantage of the long holidays such as Qingming, the Dragon Boat Festival, and National Day as an opportunity to launch new models such as short-distance, intercity, community, and rural tours, the innovation capacity of tourism models can be improved to meet the current needs of different groups of people. In addition, the advantages of tourism resources in different states of Xinjiang should be further explored based on the distribution of cold-spots and hotspots and the spatial aggregation of different categories of tourism resources to create a tourism system with regional characteristics. Promoting the construction of A-class scenic spots in Xinjiang and introducing special tourism as a breakthrough to drive Xinjiang’s economic development could also be beneficial to the improvement of the tourism industry. Xinjiang is a vast area and needs to open up tourism channels to broaden the tourism market in Xinjiang, enhancing accessibility between scenic spots while contributing to the spread of local culture. The sustainable development of the tourism industry not only requires the development of precise policy measures to support new tourism industries and define preferential policies for scenic spots to promote transformation and upgrading but also requires the full improvement of regional infrastructure to create a perfect tourism corridor. The imperfection of infrastructure facilities and the existence of barriers in the management system represent the current problems that need to be solved. Therefore, the sharing of resources, innovation, and cooperation between regions could be an efficient way to promote the blossoming of the tourism industry. In addition, with the advent of the era of big data, actively enhancing the level of regional cooperation is a key aspect in the development of the tourism industry.

## 6. Conclusions

This study analyzes the spatial distribution characteristics of different categories of tourism resource points using the statistics of POI data of tourism resources in Xinjiang. Based on the analysis results, the following conclusions are obtained:

All categories of tourism resource sites have a high density of distribution in the city of Urumqi, except for the geomorphic landscape resources. Particularly, resource sites in the category of buildings and facilities are mainly dependent on the city’s urbanization, and their distribution areas are approximately the same as those of the city. The resource points of the water landscape are concentrated in northern Xinjiang, and southern Xinjiang is limited by its natural conditions, so there exists less water distribution. Biological landscape resource sites are concentrated along the Tianshan Mountains, obeying a strip-like distribution. The geological landscape resources are widely distributed in Xinjiang, mainly in northern Xinjiang, having a low-density ring-shaped distribution around the Taklamakan Desert in southern Xinjiang. The distributions of architectural landscape and entertainment and shopping sites are similar, having Urumqi as a core of the distribution around the cities in Xinjiang. The high-density areas of ruins and relics show a cross-shaped distribution and are concentrated in the western part of Xinjiang;

The distributions of the six categories of tourism resources in Xinjiang coincide with the regional orientation of Xinjiang, and all of them have a northeast-southwest orientation. According to the standard deviation ellipse analysis results, the deflection angle of the resource points of the entertainment and shopping category is larger than those of the remaining categories of resources. In addition, the directionality and centripetal force of the resource points of the water landscape resources are not obvious, while the directionality, centripetal force, and distribution range of the remaining five categories are obvious.

According to the distribution of different categories of tourist spot resources in the cities, the resource spots in the category of site ruins are evenly distributed and have many insignificant areas. The remaining categories have the cold-spot and hotspot areas with different degrees of distribution, and many hotspot areas exist in the categories of geological and architectural landscape, which are mainly concentrated in Bazhou and Ili. There are many cold-spot and hotspot areas in the biological landscape resources, which are mainly distributed in the Altai region in northern Xinjiang and the Hotan and Kezhou regions in southern Xinjiang.

This study attempts to analyze the spatial distribution pattern of things with POI data as the research object so as to provide new ideas for the analysis of the spatial pattern of tourist resources. Economic development is of great importance for the distribution of tourism resource points, so tourism points are always clustered around the areas with highly developed economies. In this paper, the traditional geospatial analysis method is used as the main research method to analyze big geographic data. However, this paper lacks the treatment of different categories of tourism resource points and does not consider the division of resources into different levels. This can cause errors in the analysis results. The special geographical location and natural conditions of Xinjiang also imposed limitations on the study results, so future research will combine qualitative and quantitative analysis methods to overcome this limitation.

## Figures and Tables

**Figure 1 ijerph-19-07666-f001:**
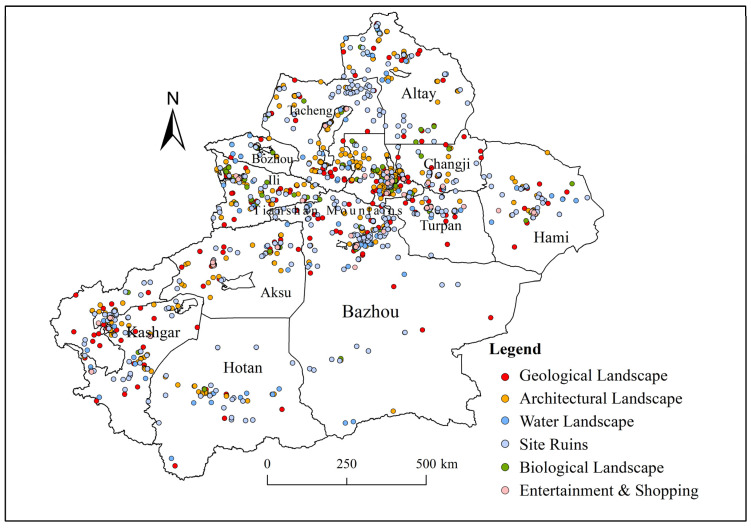
Distribution of tourism resource categories in Xinjiang.

**Figure 2 ijerph-19-07666-f002:**
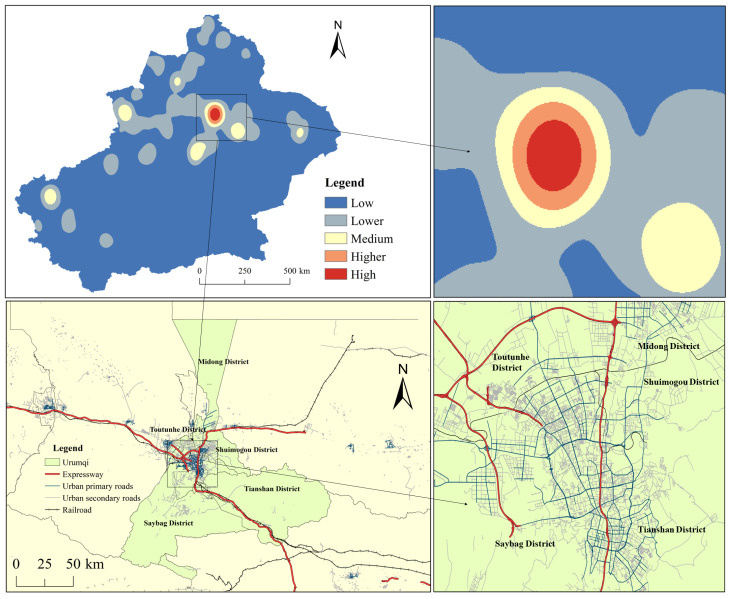
Typical distribution areas of tourism resources in Xinjiang.

**Figure 3 ijerph-19-07666-f003:**
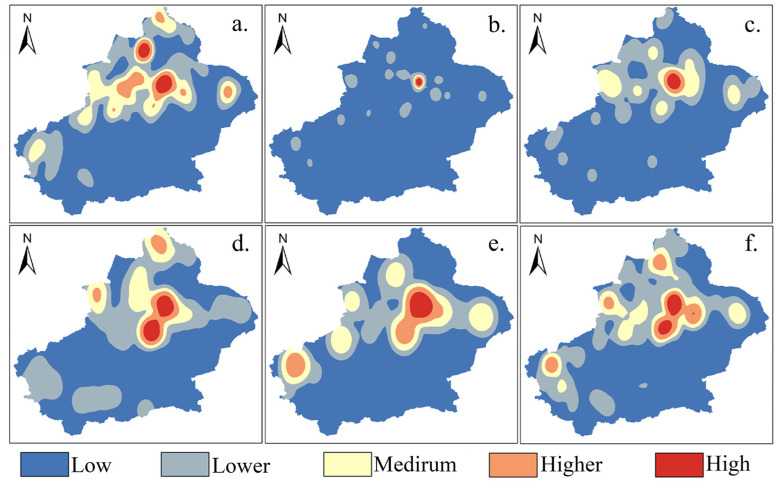
Kernel densities of different tourism resource categories in Xinjiang. (**a**) Geological landscape; (**b**) architectural landscape; (**c**) biological landscape; (**d**) water landscape; (**e**) entertainment and shopping; (**f**) site ruins.

**Figure 4 ijerph-19-07666-f004:**
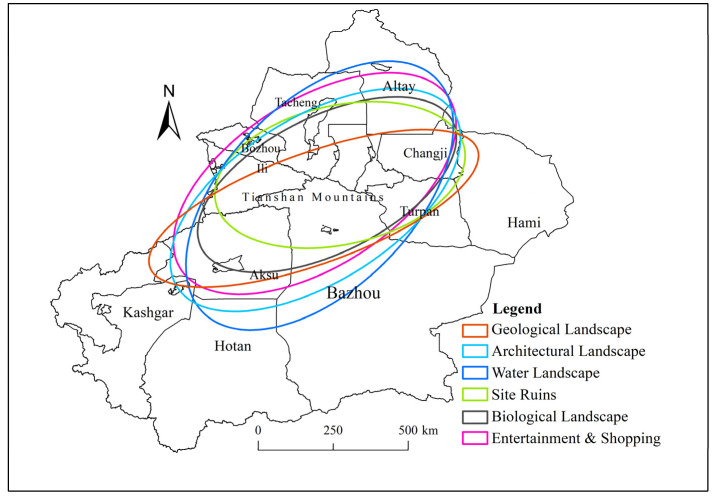
Standard deviation ellipses of different tourism resource categories in Xinjiang.

**Figure 5 ijerph-19-07666-f005:**
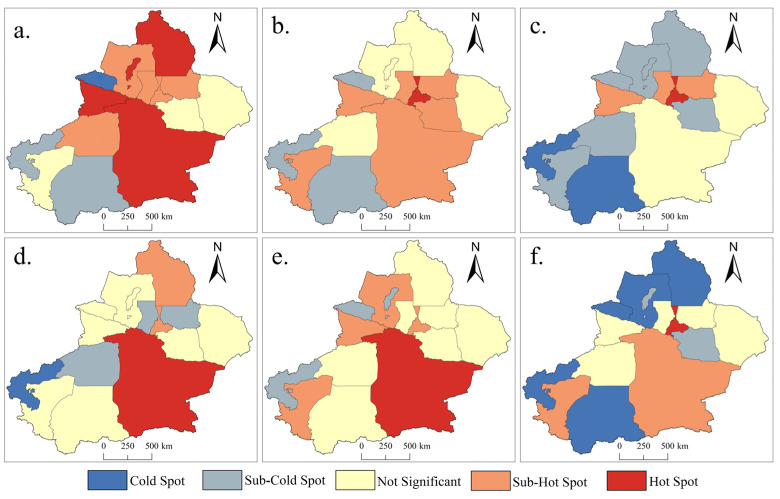
Analysis results of cold-spots and hotspots of different tourism resource categories in Xinjiang. (**a**) Geological landscape; (**b**) architectural landscape; (**c**) biological landscape; (**d**) water landscape; (**e**) entertainment and shopping; (**f**) site ruins.

**Figure 6 ijerph-19-07666-f006:**
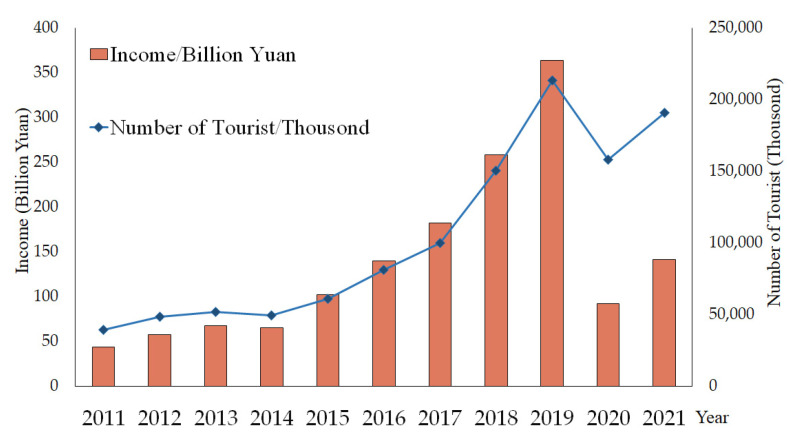
Tourism income and the number of tourists in Xinjiang from 2011 to 2021.

**Table 1 ijerph-19-07666-t001:** POI number and classification of tourism resources in Xinjiang.

Category	Samples	Content
Geological landscape	278	Mountains, valleys, and beaches
Architectural landscape	1123	Religious places of worship, cultural activities, the former residences of famous people, and historical memorial buildings
Water landscape	121	Rivers, lakes, waterfalls, and springs
Biological landscape	93	Forests, grasslands, wetlands, and animal habitats
Site ruins	464	Human activity sites, military sites, and cultural relic scattering sites
Entertainment and shopping	60	Amusement parks and water parks

**Table 2 ijerph-19-07666-t002:** Classified statistics of tourism resources in Xinjiang.

City	Tourism Resource Classification					
	Geological Landscape	Architectural Landscape	Biological Landscape	Water Landscape	Entertainment and Shopping	Site Ruins
Urumqi	23	283	18	9	10	30
Hami	17	40	8	5	5	27
Turpan	14	74	3	6	4	34
Changji	25	164	15	6	6	42
Karamay	10	48	2	6	2	4
Bozhou	3	10	1	4	2	3
Kezhou	9	32	1	2	2	18
Bazhou	34	78	9	28	8	98
Ili	34	115	18	9	5	34
Tacheng	23	82	5	7	1	49
Altay	47	46	7	17	2	34
Kashgar	18	56	2	7	7	36
Hotan	7	21	1	9	2	21
Aksu	14	74	3	6	4	34
Total	278	1145	101	127	63	468

**Table 3 ijerph-19-07666-t003:** The average nearest neighbor index results of the tourism resource categories in Xinjiang.

Category	*ANN*	Average Observation Distance (m)	Average Expected Distance (m)	*Z*-Score	*p*
Geological landscape	0.617	25,559	41,828	−12.451	0.000
Architectural landscape	0.209	4377	20,902	−49.312	0.000
Water landscape	0.467	29,281	62,718	−10.937	0.000
Site ruins	0.499	15,592	31,278	−20.238	0.008
Biological landscape	0.636	40,618	63,830	−6.526	0.000
Entertainment and shopping	0.45	28,935	64,366	−7.81	0.062

(Note: *p* < 0.05 is considered statistically significant).

**Table 4 ijerph-19-07666-t004:** Standard deviation ellipse analysis results.

Category	X-Axis Length (km)	Y-Axis Length (km)	Deflection Angle θ (°)	Flat Rate
Geological landscape	729.323	1171.974	67.627	0.37767
Biological landscape	613.065	941.945	81.578	0.34915
Site ruins	710.672	1193.733	67.388	0.40466
Architectural landscape	642.907	1040.795	71.296	0.38229
Entertainment and shopping	555.247	1239.739	75.158	0.55213
Water landscape	775.329	1206.897	58.136	0.35759

## Data Availability

Data is contained within the article.

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
