# Peer review of "Spatial Pattern Analysis of Xinjiang Tourism Resources Based on Electronic Map Points of Interest"

_ijerph, 2022, doi:10.3390/ijerph19137666_

Round 1
Reviewer 1 Report
Why is the study meaningful to tourism development? please clarify or add a statement that explains this in line 29 in the Abstract.
The policy recommendation, The future development direction of the Xinjiang tourism industry should follow the perspective of the development of holistic tourism destinations...please add clarification on a statement about holistic tourism....Overall the paper is intriguing and the research result is very interesting, I think will contribute to the journal readers' knowledge about the use of spatial analysis in tourism study.
Reviewer 3 Report
Dear authors,
First of all, congratulations for your work. However, manuscript should be modified following considerations described as follows:
Line 11. Please do not use acronyms in abstract. Please also state clearly the research’s aim.
Line 31. Please avoid use acronyms in keywords. Consider include the two version (acronym and complete).
Line 54. Please put the complete version of the acronym first time it appears.
Line 70. Author’s initial cannot be used in citations. Remove “S.”
Line 89. Delete first six words of the sentence. Redundant.
Lines 99-102. Please state clearly the research’s aim.
Line 103. Add a literature review section. You can use part of the introduction section.
Lines 105-118. Add citation of the source of the info appeared in paragraph and Figure 1.
Lines 143-177. Please always cite exact equation you are referring to.
Line 199. The paragraph which first mention Table 2 must be before its appearance and not after. Please change the order of paragraph and Table 2.
Line 241. Please put Figure 2 after the first paragraph of 3.2 Section.
Line 258. Please Add at least three decimals in Table 3 Ps.
Line 348-359. Please cite Table 4.
Line 447. Please put Figure 6 after the first paragraph of 4.2 Section. Also add source citation of Figure 6.
Line 518. Section 5.2 should be switch into conclusions, as last paragraph. You must also reinforce this paragraph, adding the limitation of the study is only circumscribed to Xinjiang, and hardly extrapolated to other parts of the world. Authors should improve the practical implications of the study to other similar regions in the world.
Line 530. Please do not numerated conclusions, display them as paragraphs.
Line 600. Ref. 11 authors name left please check.
Please do not use capital letters in journals’ names as in ref. 13, 32 and 43.
Reviewer 4 Report
A well-written article with an interesting use of GIS tools for geographical analysis of the distribution of tourist attractions. I have only formal comments:
I recommend considering the rounding of statistical data with regard to the accuracy of finding these data.
it is not clear what they are scenic spots AAA: l.64: temporal evolution characteristics of scenic spots AAA and above
first names are not used in resource references; in addition, it is missing to add et al .: l.70 Park S. [19] used big data mining
probably supposed to be "in the Greek islands": l.75-76: Dimelli [22] studied the 75 economic effects of tourism using a Greek island as a center of research
the article does not introduce what is meant by 4A and 5A tourist attraction: According to the Xinjiang Department of Culture and Tourism, since May 2021, the autonomous region has had 14 5A tourist attractions and 107 4A tourist attractions
there should be a link to the source describing the document in the source list: on the classification criteria in “Classification, Survey and Evaluation of Tourism Resources 131 (GB/T 18972-2017)”
there should be references to the specific sources from which the data were taken; the same applies to Table 1 and Table 2: The data on the number and distribution of national scenic spots, tourism revenue, and the number of visitors in different years were obtained from the Xinjiang Department of Culture and Tourism (www.xinjiangtour.gov.cn). In addition, the vector boundaries of administrative regions in Xinjiang used in this study were obtained from the latest National Basic Geographic Information System database (www.ngcc.cn).
Table 1. POI number and classification of tourism resources in Xinjiang.
Table 2. Classified statistics of tourism resources in Xinjiang.
xxxxxxxxxxxxxxxxxxxxxxxxxxxxxxxxxxxxxxxxxxxxxxxxxxxxxxxxxxxxxxxxxxxxxx
the x-axis label as well as the Y-axis unit must be added: Figure 6. Tourism income and number of tourists in Xinjiang from 2011 to 2021.
in the list of sources - there is a colon behind the "doi" and a hard space behind it, e.g.: Holik, A. Relationship of economic growth with tourism sector. JEJAK: Jurnal Ekonomi dan Kebijakan 2016, 9, 16-33, doi:https://sci-hub.ee/10.15294/jejak.v9i1.6652
Round 2
Reviewer 3 Report
Dear authors,
Manuscript should be modified following considerations not previously satisfied described as follows:
Line 116. Delete first six words of the sentence. Redundant. Not satisfied.
Line 151. Citation in Figure 1 is not correct. It should follow the MDPI citation and references policy.
Lines 180-214. Please always cite exact equation you are referring to. Not satisfied. As equations are numbered (1), (2), (3), …, (7), please cite them in text body where you are referring to.
Line 237. The paragraph which first mention Table 2 must be before its appearance and not after. Please change the order of paragraph and Table 2. Not satisfied.
Line 289. Please put Figure 2 after the first paragraph of 3.2 Section. Not satisfied.
Line 372. Table 4 must appear before Figure 4 then.
Line 506. Citation in Figure 6 is not correct. It should follow the MDPI citation and references policy.
